# Tumor-Associated Macrophages and Related Myelomonocytic Cells in the Tumor Microenvironment of Multiple Myeloma

**DOI:** 10.3390/cancers14225654

**Published:** 2022-11-17

**Authors:** Samuel S. Y. Wang, Wee Joo Chng, Haiyan Liu, Sanjay de Mel

**Affiliations:** 1Department of Rheumatology, Allergy and Immunology, Tan Tock Seng Hospital, Singapore 308433, Singapore; 2Department of Haematology-Oncology, National University Cancer Institute Singapore, National University Health System, Singapore 119228, Singapore; 3Department of Medicine, Yong Loo Lin School of Medicine, National University of Singapore, 10 Medical Dr, Singapore 117597, Singapore; 4Cancer Science Institute, National University of Singapore, 14 Medical Dr, #12-01 Centre for Translational Medicine, Singapore 117599, Singapore; 5Immunology Programme, Life Sciences Institute, National University of Singapore, Singapore 117456, Singapore; 6Immunology Translational Research Program, Department of Microbiology and Immunology, Yong Loo Lin School of Medicine, National University of Singapore, Singapore 117456, Singapore

**Keywords:** multiple myeloma, tumor-associated macrophages, myeloid derived suppressor cells, dendritic cells, tumor microenvironment

## Abstract

**Simple Summary:**

Multiple myeloma (MM) is the second-most common blood cancer and is currently incurable despite recent advances in treatment. The immune cells which are present in the vicinity of the MM tumor cells comprise the tumor immune microenvironment. The tumor microenvironment of MM patients is thought to play an important part in how they respond to treatment. It is hypothesized that dysfunction of immune cells in MM patients may result in resistance to treatment. Tumor-associated macrophages (TAM), myeloid-derived suppressor cells (MDSC), and dendritic cells (DC) are important components of the tumor microenvironment in MM. This review aims to provide an overview of the biology and clinical relevance of TAMs, MDSCs and DCs in the MM immune microenvironment. We will also provide our perspective on how novel technologies can be applied to studying these cells and how they may impact treatment strategies of the future.

**Abstract:**

Multiple myeloma (MM) is the second-most common hematologic malignancy and remains incurable despite potent plasma cell directed therapeutics. The tumor microenvironment (TME) is a key player in the pathogenesis and progression of MM and is an active focus of research with a view to targeting immune dysregulation. Tumor-associated macrophages (TAM), myeloid derived suppressor cells (MDSC), and dendritic cells (DC) are known to drive progression and treatment resistance in many cancers. They have also been shown to promote MM progression and immune suppression in vitro, and there is growing evidence of their impact on clinical outcomes. The heterogeneity and functional characteristics of myelomonocytic cells in MM are being unraveled through high-dimensional immune profiling techniques. We are also beginning to understand how they may affect and be modulated by current and future MM therapeutics. In this review, we provide an overview of the biology and clinical relevance of TAMs, MDSCs, and DCs in the MM TME. We also highlight key areas to be addressed in future research as well as our perspectives on how the myelomonocytic compartment of the TME may influence therapeutic strategies of the future.

## 1. Introduction

Multiple myeloma (MM) is the second-most common hematologic neoplasm worldwide [1,2]. Although advances in therapeutics have led to improved survival, all patients eventually develop treatment resistance and succumb to this disease [3]. While much research has focused on targeting clonal plasma cells (PC) in MM, the tumor microenvironment (TME) has recently become a focus of interest [4]. Immune dysregulation is a hallmark of cancer and has also been implicated in treatment resistance and progression of MM [2,5]. MM PCs interact with numerous components of the bone marrow (BM) TME to support their survival and proliferation [2]. While natural killer (NK) cells and T cells are known to be dysfunctional in MM, PCs also interact with the BM stroma as well as cells of the myelomonocytic lineage [6,7,8]. 

Macrophages are best known for their phagocytic and antigen-presenting functions but are also implicated in malignancy [9]. They are a highly heterogeneous cell type defined by surface expression of CD68 and CD14 but can be subclassified based on CD163 or CD206 expression, among other aspects [9]. Tumor-associated macrophages (TAMs) have been a field of active research in cancer immunology for decades [10,11]. In addition to phagocytosis of cancer cells, macrophages are responsible for modulating T- and NK-cell-mediated anti-tumor responses [10,12,13]. Specifically, they can influence CD8 T cell proliferation and recruit T-regulatory cells, resulting in impaired anti-tumor immune responses [13]. There is likely to be a complex interplay between TAMs, tumor cells other immune cells in the TME which maybe unique to each malignancy [14]. 

TAMs are known to drive tumor progression in a variety of solid cancers (summarized in Table 1) [15,16]. The interaction between TAMs and cancer stem cells appears to be particularly important in this context, with TAMs promoting the survival and self-renewal of cancer stem cells via a variety of growth factors [15,17]. Cancer stem cells, in turn, educate TAMs to enhance the stem-like properties of tumor cells [18]. It is noteworthy that the WNT signaling pathway was implicated in this crosstalk, [18] suggesting the potential for therapeutic targeting [19]. TAMs also promote tumor metastasis via a variety of mechanisms, including angiogenesis, regulation of the epithelial mesenchymal transition, and education of mesenchymal stromal cells [20,21]. It has been demonstrated that transfer of microRNA via exosomes is an important mechanism mediating the crosstalk between tumor cells, TAMs, and stromal cells. This represents an important field for future research and maybe a potential therapeutic target [22,23].

TAMs were traditionally dichotomized into anti-tumor (M1) or pro-tumor (M2) subtypes and were thought to arise from circulating monocytes [12]. Recent evidence suggests that this characterization maybe an oversimplification given that TAMs may have dynamic roles influenced by the tumor they reside in [10,24]. We have also made great strides in our understanding of how TAMs interact with tumor cells. CD47 expressed on cancer cells interacts with its ligand signal regulatory protein alpha (SIRPα) on macrophages [25]. The CD47-SIRPα interaction generates a “don’t eat me” signal, protecting cancer cells from macrophage mediated phagocytosis [26]. 

Multiple subsets of TAMs have recently been demonstrated in hepatocellular carcinoma (HCC) via single-cell RNA sequencing and high-dimensional flow cytometry [27]. Intriguingly, folate-receptor-2-expressing TAMs exhibited a phenotype similar to that of fetal macrophages suggesting an “onco-foetal reprogramming” of TAMs in HCC. These “onco-foetal” macrophages were also shown to have immunosuppressive effect on T cells. High-dimensional profiling of TAMs may reveal unique subsets, phenotypes, and functions of these cells in other tumors, including MM. The heterogeneity of TAMs across tumor types (Table 1) highlights the importance of deep immune profiling of the TME to definitively evaluate TAM subsets and their function in each cancer. 

Myeloid-derived suppressor cells (MDSC) are a heterogeneous population of immature myeloid cells that are usually absent in healthy individuals but have been recently implicated in cancer [8]. They are immunophenotypically defined by the expression of CD11b, CD33 and negativity or low expression of HLA-DR [8]. MDSCs arise from hematopoietic progenitor cells and are broadly divided into granulocytic (CD15+, CD14-) and monocytic (CD15-, CD14+) subtypes [8,40]. An immature subset of MDSCs, which express CD33 and CD11b but lack HLA-DR, has also been identified and termed early-stage MDSCs (E-MDSCs) [41]. The expansion of MDSCs maybe driven by colony stimulating factors followed by activation through signal transducer and activator of transcription 3 (STAT3) and nuclear factor κB signaling [40]. MDSCs dysregulate anti-tumor responses mediated by T-cells, NK cells, and macrophages. Depletion of arginine in the TME and inhibition of T cell receptor nitrosylation are mechanisms by which MDSCs affect anti-tumor T cell responses while they also promote immunosuppressive polarization of TAMs [14]. Besides immune dysregulation, MDSCs also promote angiogenesis and metastases, adding to their tumor enhancing potential [14,42,43].

Unlike MDSCs, dendritic cells (DC)s are a vital component of the normal immune system and play a key role in antigen presentation to naïve T cells via major histocompatibility complex (MHC) molecules [44]. They comprise conventional dendritic cells (cDC) and plasmacytoid dendritic cells (pDC) which are immunophenotypically and functionally distinct [45,46]. cDCs are potent antigen-presenting cells and have the ability to harness anti-tumor T cell immunity, but these responses can be blunted by tumor-derived cytokines and regulatory T cells [47]. pDCs are known to induce tolerogenic signals in the TME and are associated with an adverse prognosis in many cancers [48]. Specific receptors such as chemokine receptor 5 (CCR5) aid immature DCs in migrating to sites of inflammation, while chemokine receptor 7 (CCR7) aids mature DCs in migrating to secondary lymph nodes. Reduced expression of lymphocyte antigen 75 (DEC-205) decreases antigen uptake by DCs [48]. 

TAMs, MDSCs, and DCs have recently emerged as important players in the MM TME. In this review, we will discuss current concepts on how these cells contribute to immune dysregulation and tumor progression in MM. We also elaborate on potential mechanisms of treatment resistance brought about by these cells and how they may be overcome. Lastly, we explore limitations of our knowledge on their biology and how future research can address these gaps. 

## 2. The Role of Tumor-Associated Macrophages in the Pathogenesis and Progression of Myeloma

In vitro data have demonstrated that TAMs support PC growth and impair anti-tumor immunity in MM [49,50]. They are more abundant in MM compared to monoclonal gammopathy of unknown significance (MGUS), suggesting a role in the progression from MGUS to MM [51]. It has also been proposed that TAMs recruit peripheral blood monocytes into the TME through their interaction with BM stromal cells [51]. TAMs promote angiogenesis in MM through vascular mimicry and secretion of vascular endothelial growth factor (VEGF) as well as matrix metalloproteases, synergizing with the MM cells’ own angiogenic properties [52,53,54]. This is supported by the finding of microvessel density correlating with TAM infiltration based on immunohistochemistry (IHC) in BM trephine samples [39]. The polycomb group protein BMI1 was recently shown in murine models to modulate the pro-myeloma functions of TAMs. TAMs showed higher BMI1 levels compared to normal macrophages, and BMI1 knockdown reduced their angiogenic potential. These data suggest that the angiogenic functions of TAMs may depend on multiple, overlapping pathways. Future research should focus on identifying clinically actionable targets in this context [9]. 

There also seems to be an overlap between the pro-angiogenic capability of TAMs and their ability to modulate immunosuppression. A study investigating exosome-derived miR-let-7c identified its association with driving TAM polarization to the immunosuppressive phenotype and promoting angiogenesis in the TME [55]. MM cells have also been shown to drive immunosuppressive polarization of macrophages in vitro, resulting in elevated expression of the M2-related scavenger receptor CD206 and reduced lipopolysaccharide-induced TNFα secretion, which is a hallmark of the pro-inflammatory macrophage response [51]. Macrophages cocultured with MM cells suppressed T cell proliferation and IFN-γ production in response to T cell receptor activation, providing further support for their ability to educate TAMs [51]. Coculture with TAMs also increased programmed death receptor 1 and reduced granzyme B expression on CD8 T cells and increased programmed death receptor ligand-1 expression on MM cells, suggesting that they drive immune tolerance [9]. 

In addition to suppressing anti-tumor immunity, TAMs have also been shown to promote chemoresistance in MM. Zheng et al. demonstrated that macrophages cocultured with MM cells protected them from melphalan-mediated toxicity through the interaction between P-selectin and ICAM-1 expressed on the TAMS and P-selectin glycoprotein ligand 1 and CD18 on the MM cells [50,56]. MM cells influenced by macrophages were found upregulate phosphorylated Src and Erk1/2 in addition to MYC, suggesting that these signaling pathways were driving the treatment resistance [56]. Coculture studies have also demonstrated crosstalk between TAMs and mesenchymal stem cells leading to impaired bortezomib mediated apoptosis through IL-6 and IL-10 [57]. The Stat3 pathway has also been implicated in TAM-mediated chemoresistance and was overcome in vitro using Janus kinase (JAK)2 inhibition [49]. JAK inhibition was also shown to suppress immunosuppressive TAM polarization and overcome resistance to lenalidomide induced by coculture of TAMs with MM cells [58]. Taken together, these data indicate that TAMs play a crucial role in the pathogenesis of MM as well as chemoresistance. As many of the underlying pathways are potential clinical targets, they are an important focus for future translational research. It is noteworthy, however, that drug resistance in MM is a complex, multifactorial process [59]. Our understanding of the role of TAMs in treatment resistance is likely incomplete given our limited knowledge of TAM subsets and their role in the MM TME. The known interactions between MM plasma cells, and TAMs are summarized in Figure 1.

## 3. The Clinical Relevance of Tumor-Associated Macrophages in Multiple Myeloma

TAM infiltration of the BM assessed using IHC has been proposed as an adverse prognostic factor in MM [38,39,60,61,62,63,64]. Wang et al. studied a cohort of patients treated with bortezomib-based regimens and defined TAMs based on CD163 expression. They used 55 macrophages per high-power field to define high TAM infiltration and found that these patients had more aggressive clinical presentations, lower rates of complete response, and inferior progression-free (PFS) and overall survival (OS) [62]. In a similar analysis, Chen et al. demonstrated that CD163 combined with inducible nitric oxide synthetase (iNOS) identified a subset of TAMs which were an independent prognostic factor in a cohort treated with proteasome inhibitors or immunomodulators [60]. These findings were replicated in separate studies using CD68 as a single marker [63] or CD68 and CD163 together to define TAMs [39]. 

CD163 was also used to identify TAMs based on IHC and flow cytometry in a cohort representing the spectrum of plasma cell dyscrasias (MGUS, smoldering myeloma, and newly diagnosed and relapsed MM) [38]. This study also showed that TAM infiltration had an adverse effect on OS and in addition demonstrated that the matrix metalloproteinase (MMP) inducer CD147 was expressed concurrently with CD163 and also correlated negatively with OS [38]. As MMPs are known to be one of the mechanisms by which TAMs promote angiogenesis, it is possible that CD147 could mediate this. 

Wu et al. studied the impact of different TAM subsets identified by IHC on clinical outcomes. They identified macrophages based on CD68 expression and subclassified them as classically activated (M1) by iNOS expression or alternatively activated (M2) by CD163 expression [61]. They demonstrated that patients with a high M2 infiltration had a significantly inferior PFS and OS compared to those with a high M1 infiltration. This finding was reinforced by Beyar-Katz et al., who reported low M1 (defined by CD68 and CCR2 expression by flow cytometry) infiltration to be associated with inferior responses to bortezomib [64]. Although the limited antigen panels used in these studies may not accurately identify TAM subsets as we understand them today, these data give us a glimpse of the heterogeneity and clinical impact of TAM subpopulations in the MM TME. 

Given the heterogeneity of TAMs, defining an optimal antigen panel to identify them maybe a challenge. In a recent study evaluating TAMs in HCC, macrophages were identified based on their CD45+/CD68+/CD14+ phenotype and subclassified based on their expression of CD163 and CD206 into TAM1 and TAM2 subsets among others [27]. The expression of CD206 and CD163 along with IL-10, STAT-3, VEGF and MMP2/9 has been proposed as characteristic of immunosuppressive TAMs [65,66]. High-dimensional profiling of TAMs across multiple tumor subtypes may be necessary in order to arrive at a more “definitive” characterization encompassing the many different subsets that are likely to exist. 

Circulating biomarkers that may reflect TAM infiltration have been actively studied, with soluble CD163 and CD206 being particularly promising [67]. Indeed, CD206 levels have been shown to correlate with response to treatment and was even reported as an independent prognostic factor for OS [67]. These studies together with the IHC data described above provide strong support for the clinical relevance of TAMs in MM and justify further translational research to unravel the underlying mechanisms. A better understanding of the TAM–plasma cell interaction may provide the basis for targeted therapeutics to overcome TAM-mediated treatment resistance. For example, cyclophosphamide was found to potentiate daratumumab-mediated killing of MM cells by altering the TME to promote macrophage recruitment, polarization to a proinflammatory phenotype, and directing antibody-dependent cellular phagocytosis [68]. Table 2 summarizes the studies demonstrating the clinical importance of TAMs in MM. 

## 4. The CD47: SIRPα “Macrophage Checkpoint” in Multiple Myeloma

CD47 (formerly known as integrin-associated protein) is expressed on a variety of cell types and plays a physiological role in immune tolerance [69]. Specifically, it generates a “don’t eat me” signal through its interaction with its ligand—signal regulatory proteinα (SIRPα)—which is expressed on macrophages [25,69]. CD47 has also become a focus of research in cancer, as tumor cells express CD47 to escape macrophage-mediated phagocytosis [25]. CD47 is expressed on MM PCs, but not normal PCs, and is associated with progression of MGUS to MM [25,54,70,71]. CD47 expression quantified by IHC was associated with adverse outcomes in MM patients treated with vincristine doxorubicin dexamethasone, highlighting the clinical importance of this checkpoint [72]. It is also noteworthy that CD47 expression by flow cytometry was less prominent in extramedullary MM compared to BM samples taken from the same patients [71].These data suggest that the macrophage checkpoint maybe more important in BM disease than in extramedullary MM, highlighting the importance of spatial tumor heterogeneity. 

More recently, Kambhapathi et al. studied the role of TAMs/CD47 expression in the response to the CD38 monoclonal antibody daratumumab in relapsed MM [73]. They showed that TAM infiltration by IHC remained unchanged when comparing BM biopsies performed before daratumumab treatment and at relapse. Interestingly, they demonstrated that CD47 expression changed from surface to cytoplasmic at relapse, but the total expression level was constant. The expression of CD47 on plasma cells should be evaluated at the single-cell level using higher-resolution techniques to provide insights into the biological implications of this finding. The role of the “macrophage checkpoint” in MM was further supported by experiments, demonstrating that CD16+ monocytes are required for anti-CD47-antibody-mediated killing of MM plasma cells [74]. 

Targeting the CD47-SIRPα interaction to inhibit the “macrophage checkpoint” is being actively evaluated as a therapeutic strategy [70,74,75,76,77]. The best-characterized agents are anti-CD47 antibodies, which have proven effective in inducing phagocytosis of cancer cells in vitro as well as inhibiting tumor growth in murine models of both hematologic and solid tumors [75,78]. Indeed, this strategy has shown clinical efficacy in B-cell lymphomas when used in combination with CD20 monoclonal antibodies [79]. CD47 antibodies have recently been explored against MM cell lines, where they enhanced macrophage-induced plasma cell phagocytosis [54]. Other therapeutic strategies showing promise include CD47 peptide agonists as well as bispecific antibodies and liposomal encapsulated micro-RNAs [25]. 

A number of ongoing clinical trials are evaluating CD47-directed therapy in MM (NCT04445701, NCT05139225, NCT04892446, NCT02663518). These agents are being evaluated as monotherapies as well as in combination with existing MM therapeutics [25]. The results of these studies are eagerly awaited, and translational research should focus on delineating the immune profile of responders and non-responders with a view to developing personalized therapeutic strategies. An overview of the studies evaluating the CD47:SIRPα checkpoint in MM is presented in Table 3. 

## 5. Myeloid-Derived Suppressor Cells in Multiple Myeloma

MDSCs are proposed to influence innate and adaptive immune responses in MM, resulting in tumor progression [14]. Growth factors produced by the MM TME have been shown to impair myeloid differentiation in murine models [14]. Among these, granulocyte-macrophage colony-stimulating-factor, granulocyte-colony-stimulating-factor, monocyte-colony-stimulating-factor, stem cell factor, VEGF and interleukin-3 (IL-3) promote the reprogramming of immature myeloid cells into immunosuppressive MDSCs and recruit them into the TME [14]. The pro-inflammatory milieu of the MM TME also leads to abnormal differentiation and localization of MDSCs via cytokines in addition to growth factors [80]. Interleukin-18 was shown to be particularly effective at generating MDSCs in murine models, where it also promoted MM tumor progression [81].

MDSCs have been demonstrated in the BM of MM patients at significantly higher levels compared to normal controls [82]. It is noteworthy that granulocytic MDSC (G-MDSC) defined by CD11b+, CD14-, CD33+, CD15+ expression were more prominent in MM patients [82], while monocytic MDSC (M-MDSC) with a CD11b+, CD14+, CD33+, CD15- phenotype had a similar prevalence in MM patients and controls [82]. More recent studies showed that G-MDSC themselves comprise several subsets, including neutrophils with distinct maturation stages, eosinophils, and basophils [8]. Among these, only the mature neutrophil subset was more prevalent in MM patients compared to healthy controls. Indeed, these neutrophils showed upregulation of G-MDSC genes, suggesting they are a unique subtype of G-MDSC found in MM, and can be identified using the CD11b+, CD13+, CD16+ phenotype. E-MDSCs have also been detected in the PB of MM patients and increased in number after induction therapy, in contrast to M-MDSCs, which decreased in number [83]. Taken together, these data suggest that MDSCs in MM may be distinct from those in other malignancies and may comprise a variety of subsets with distinct functions in the TME. 

The impact of MDSC on the anti-MM cytotoxic T cell response was demonstrated in S100A9 knockout (KO) mice, which have impaired MDSC accumulation [82]. The S100A9 KO mice showed an accumulation of antigen-specific CD8+ T cells in the BM, which correlated with reduced MM cell proliferation [82]. M-MDSCs and E-MDSCs obtained from MM patients pre-autologous stem cell transplant (ASCT) were both able to suppress T and NK cell proliferation in vitro [83]. Interestingly, post-ASCT MDSCs of both subtypes had lost their ability to suppress T cell responses. Pre-ASCT M-MDSCs also strongly inhibited the in vitro cytotoxic effect of melphalan in contrast to E-MDSCs [83]. Colony-stimulating factor 1 receptor (CSF1R) was proposed to mediate the protective effect of MDSCs, as blockade of CSF1R restored melphalan-induced cytotoxicity which was impaired by pre-ASCT MDSC [83]. The negative impact of IL-18-driven MDSCs on T cell function was also shown in the VκMYC murine model. These data collectively show that MDSCs inhibit T cell responses via a variety of mechanisms and play a critical role in the dysregulated MM TME. 

Beyond their role as T cell suppressors, G-MDSCs also promote stemlike properties and tumorigenic potential in MM through inducing piRNA-823 expression and activation of DNMT3B [84]. Other pathways associated with MDSCs include the 5’ AMP-activated protein kinase (AMPK) pathway, which functions as a cellular energy sensor and regulates lipid and glucose metabolism [85], AMPK signaling results in increased MCL-1 and BCL-2 expression, as well as the autophagosome formation marker LC3II, which together may impair apoptosis [85].

A growing body of evidence suggests that MDSCs are linked to clinical outcomes in MM. The frequency of the “mature neutrophil subset” of G-MDSC was associated with inferior PFS in patients treated on the GEM2012MENOS65 trial [8] and OS in a separate real-world cohort [84]. In contrast, the frequency of M-MDSC appeared to inversely correlate with time to progression when they were quantified pre-ASCT [83]. These data support the hypothesis that different subsets of MDSCs are functionally distinct. Interestingly, none of the MDSC subsets post-ASCT had an impact on time to progression (TTP), indicating that high-dose melphalan and stem cell rescue have a significant impact on the TME [83]. This is an area worthy of further evaluation through high-dimensional profiling studies.

Given the strong biological evidence for MDSC-mediated immune suppression and emerging data on its clinical relevance, MDSC maybe an attractive therapeutic target [14]. The impact of current MM therapeutics such as proteasome inhibitors and immunomodulators on MDSC remain uncertain. Coculture of MM cells and monocytes with bortezomib treatment resulted in a reduced number of M-MDSCs [86], while the combination of bortezomib and lenalidomide had no effect on the immune suppressive capacity of MDSCs [87]. As proteasome inhibitor/immunomodulator combinations are widely used for MM therapy, future studies should focus on exploring their impact on MDSC populations in clinical samples. Immunotherapeutics are an emerging treatment modality for MM and are likely to have an impact on the TME [88]. It is noteworthy that while daratumumab has well-documented effects on T and NK cells in the TME, it has not yet been shown to impact MDSCs [8]. 

As M-MDSC have been shown to acquire G-MDSC characteristics through epigenetic silencing in cancer, histone-deacetylase (HDAC) inhibitors are being evaluated as a means of reversing this phenotype and driving M-MDSC to differentiate into macrophages and DCs [14,89]. Given that HDAC inhibitors also have direct toxicity against plasma cells, their role in immune modulation would be an important focus for future research. Therapeutically targeting the cytokines which drive MDSCs such as IL-18 [81] and CSF1R [83] may be another option for indirectly modulating MDSCs in the TME. Deeper knowledge of the biology of MDSCs in MM and their role in the response to current therapeutic regimens would be critical before these strategies can be taken forward into clinical trials. 

## 6. Dendritic Cells in Multiple Myeloma

DCs play a crucial role in the pathogenesis and progression of MM [48,90,91]. Interestingly, the progression from MGUS to MM is associated with an increase in both cDCs and pDCs in the BM niche [48]. The immunophenotypic profile of DCs is also significantly altered in MM patients. The expression of CCR5, CCR7, and DEC-205 was downregulated on all peripheral blood DC subtypes in MM compared to controls [48,91,92]. The downregulation of CCR5 and CCR7 impair DC migration to sites of inflammation, while reduced DEC-205 impairs antigen uptake. Taken together, the DC dysregulation in MM results in dysfunctional maturation and hampers their antigen-presenting capability, which in turn results in impaired anti-tumor T cell activation [48,90,91].

MM plasma cells also influence DC development through a cocktail of cytokines comprising transforming growth factor-β1 (TGF-β1), VEGF, IL-6, and IL-10 [48]. These immunologically inhibitory cytokines lead to disrupted DC differentiation through hyperactivation of the STAT3 and extracellular signal-regulated kinase (ERK) pathways [93,94]. These data show that MM PCs have evolved numerous mechanisms to escape normal DC-mediated tumor surveillance but also suggest that some of these may be potential therapeutic targets.

Based on the current understanding of DCs, there has been renewed interest in developing DC-specific immunotherapies in MM [48]. Generating an anti-MM immune response through DC vaccination with idiotypic (Id) protein as MM-specific tumor-associated antigen was one such therapy under investigation [48]. Initial studies demonstrated the functionality of the ex-vivo-generated DCs although clinical responses were limited [48,95]. Other approaches to eliciting an immune response from DCs include the usage of MM-associated antigen mRNA or patient-derived MM cells [48,96,97]. These therapies remain in early phase clinical trials and clinical responses are thus far limited [48]. It is likely that DC-based treatments as monotherapy may not be adequate and combinations with established anti-myeloma agents are more likely to succeed and should be evaluated in clinical trials. With a better understanding of the crosstalk between PC, MDSC, and DC (summarized in Figure 2), immunotherapies specifically targeting the interactions between these cells should be the subject of future research.

## 7. High-Dimensional Profiling of TAMs and Related Myelomonocytic Cells in Cancer

The majority of studies evaluating TAMs in MM have relied on IHC or flow cytometry with a limited marker panel to identify and describe TAMs. These studies also predominantly used a binary classification of M1 (anti-tumor) as opposed to M2 (pro-tumor) phenotypes. The last decade has brought about a rapid escalation in our knowledge of macrophage biology, especially concerning their ontogeny, heterogeneity, and plasticity [10,98]. The M1 vs. M2 classification is hence likely to be too simplistic to adequately understand TAMs, and more sophisticated techniques are needed to provide more granular insight into the phenotype of the various TAM subsets. As discussed above, MDSCs and DCs are also highly heterogeneous in phenotype and function and may not be adequately characterized by conventional techniques. 

A multiparametric approach is required to evaluate complex populations, such as myelomonocytic cells in the TME. At the protein level, cytometry by time-of-flight mass spectrometry (CyTOF) has emerged as a powerful technique allowing the simultaneous evaluation of up to 40 cell surface or cytoplasmic markers [99]. CyTOF has been successfully applied to study TAMs in lung and brain malignancies [16] among others. Spectral flow cytometry has also shown potential as a tool for high-dimensional immune profiling, overcoming the limitations of spectral overlap associated with conventional flow cytometry [100]. Single-cell RNA sequencing (scRNAseq) has great value as a tool to unravel tumor microenvironmental heterogeneity [16]. Indeed, scRNAseq has been applied to study TAMs in relapsed MM, showing not only multiple subsets but also unique transcriptional features, indicating distinct functions and interactions with other cells in the TME [101,102]. 

Information on spatial relationships of TAMs within tumors can be elicited by multiplexed immunohistochemistry or spatial transcriptomics which have also been successfully applied in other malignancies [16]. An integrated approach using a combination of these platforms may provide the most comprehensive assessment of TAMs and related myelomonocytic cells in MM. 

## 8. Reprogramming of TAMs towards an Anti-Myeloma Phenotype

Reprogramming of TAMs to augment their anti-tumor phenotype while reducing their immunosuppressive properties has been of significant interest [66]. One strategy employed a combination treatment in which a pro-M1 cytokine granulocyte-macrophage colony-stimulating factor (GM-CSF) and a pro-M2 cytokine macrophage migration inhibitory factor were used together [103]. This induced M1 genes and in vitro antitumor effects which exceeded the activity of GM-CSF alone [103]. Furthermore, the dual treatment resulted in macrophage-dependent therapeutic responses in a xenograft murine model [103]. 

Other growth-factor-related pathways of interest include the CSF1 receptor (CSF1R). Wang et al. used CSF1R-blocking mAbs to inhibit MM growth by depleting TAMs, polarizing them to the M1 phenotype, and inducing a tumor-specific CD4 +T cell response. The CSF1R-blocking mAbs together with bortezomib or melphalan displayed additive in vitro activity. These results suggest anti-CSF1R mAbs may be a potential method of repolarizing TAMs to promote anti-myeloma immunity and enhancing responses to conventional treatment [104]. The impact of these pathways on the efficacy of immunotherapy in MM is an important area to be explored in future studies. 

Other antibody targets under evaluation include CD40, which is a cell surface costimulatory protein found on macrophages in addition to other antigen-presenting cells. Antibodies activating CD40 have been observed to generate a macrophage-repolarizing effect through the ligation of toll-like receptors (TLR). This strategy has been successful at inducing immune responses against MM in both ex vivo and in vivo models [105]. IL-10 signaling may be another therapeutic target of interest. IL-10 secretion by MM cells polarizes macrophages towards an M2 phenotype which supports the proliferation of MM cells and drug resistance. Inhibition of IL-10 signaling using an antibody against the IL-10 receptor resulted in the reprogramming of TAMs towards the anti-tumor phenotype. IL-10 receptor blockade reduced MM proliferation and overcame resistance to lenalidomide and dexamethasone both in vitro and in vivo [106]. Both CD40 agonist antibodies and IL-10 blocking antibodies are already in clinical trials for cancer immunotherapy [107,108]. They are worthy of exploration in clinical trials for MM, especially in combination with other immunotherapies

Small molecule inhibitors of specific signaling pathways are also under active study in the field of reprogramming TAMs. Ruxolitinib, a well-established JAK1/2 inhibitor, was shown to suppress the pro-tumor phenotype in macrophages by reducing the expression of Tribbles homolog 1 protein kinase. Ruxolitinib promoted M1 polarization and overcame lenalidomide resistance both in vitro and in vivo [9]. JAK/STAT inhibition has shown preclinical promise in MM therapy via multiple mechanisms [109]. The immunoregulatory role of these pathways will hence be of significant interest in the quest to bring these agents into the clinic. 

Other pathways that influence TAM development in MM include the BMI1 protein, which promotes macrophage proliferation as well as angiogenesis, drug resistance, and proliferation of MM cells. BMI1 upregulation appears to occur via the hedgehog–myc axis and sonic hedgehog secretion by MM cells was identified to be critical for this pathway. The BMI1 inhibitor PTC596 decreased tumor burden and improved the survival of mice in a murine myeloma model by depleting pro-tumor TAMs [9]. Further studies are required to evaluate the potential clinical applications of BMI inhibition. 

## 9. The Role of Tumor-Associated Macrophages, Dendritic Cells, and Myeloid-Derived Suppressor Cells in the Era of Immunotherapy- and Immunomodulator-Based Treatment of Multiple Myeloma

The advent of immunotherapy has revolutionized the treatment of MM. Daratumumab (Dara) and isatuximab are humanized monoclonal antibodies (mAbs) against CD38, which is strongly expressed on PCs [110,111]. CD38 mAbs induce the killing of PCs via a variety of mechanisms, including macrophage-mediated antibody-dependent cellular phagocytosis (ADCP) [110]. Despite their efficacy, all patients eventually develop resistance to CD38 mAbs, and some may have suboptimal responses [88]. This is particularly true for patients with high cytogenetic risk MM, who may not benefit as much from induction treatment with dara as standard risk patients [112]. While the impact of anti-CD38 mAbs on T cell and NK cell populations has been described, the role of TAMs, MDSCs, and DCs in the response to these agents remains an important knowledge gap [113,114,115,116,117]. Mass cytometric analysis of the BM in relapsed/refractory MM patients treated with Dara plus pomalidomide and dexamethasone showed an increase in total monocytes in responders but not in non-responders (45). Interrogation of macrophage/monocyte subsets and identification of TAMs were not, however, reported in this study.

Immunomodulators (IMIDs) have been part of MM treatment protocols for decades, with thalidomide being the first in this class to show efficacy [118]. They have pleiotropic anti-MM properties, including immune modulation, anti-angiogenic, anti-inflammatory, and cytotoxic/anti-proliferative effects [119]. Lenalidomide has been shown to drive TAMs towards a pro-inflammatory/anti-tumor phenotype by modulating the CRBN-CRL4 E3 ligase to ubiquitinate and degrade the transcription factor IKAROS family zinc finger 1(IKZF1) [120]. The angiogenic properties of TAMs are also countered by IMIDs, which have anti-angiogenic properties exerted at least partly through modulation of VEGF and TNF alpha [119]. IMIDS are known to synergize with mAbs, and these combinations are now in routine use for newly diagnosed MM [85,121,122,123]. The impact of combining mAbs with IMIDS on TAMS and related myelomonocytic cells remains an important unanswered question. 

There is comparatively little data on the role of MDSCs and DCs in the era of immunotherapy, with some studies suggesting that dara treatment may not impact G-MDSC populations [8]. Future studies should seek to evaluate TAMs, MDSCs, and DCs in patients treated with mAb/IMID combinations to identify pathways responsible for treatment resistance in this setting. The role of TAMs, MDSCs, and DCs in the response to novel immunotherapeutics, such as chimeric antigen receptor T cells, Bi specific T cell engagers, and antibody-drug conjugates, is an exciting field that is worthy of evaluation in future studies. Despite the efficacy of these agents, we remain uncertain on how best to sequence them and in which patients groups they would be most effective. 

Personalized therapeutics in MM may be achieved through ex vivo drug sensitivity testing, which has shown promise in a variety of hematologic malignancies [124,125,126]. An important limitation of most ex vivo drug testing platforms is the absence of a tumor microenvironment, resulting in failure to accurately reflect the impact of treatment in vivo. This is particularly true for immunotherapy which often rely on the TME to exert their anti-tumor effects [88].The development of three dimensional tumor models incorporating TAMs and stromal cells is an important step towards achieving this goal and is certainly worthy of exploration in MM [127,128].

## 10. Conclusions

In this review, we have highlighted the importance of TAMs and related myelomonocytic cells in the pathogenesis and clinical outcomes of MM. The emergence of the CD47-SIRPα axis as a key immunologic checkpoint raises the promise for effective novel therapies. The TME will play an increasingly important role in the era of immunotherapy for MM, with TAMs, MDSCs, and DCs likely to affect responses to many novel immunotherapeutics. As we learn more about the complex biology of the MM TME and the role of myelomonocytic cells, the need for high-dimensional profiling techniques is becoming apparent. We propose that translational research on myelomonocytic cells will require an integrated approach nested in clinical trials of immunotherapy-treated patients. This may allow a truly personalized application of immunotherapy for MM, which may be a key step in our quest for a cure. 

## Figures and Tables

**Figure 1 cancers-14-05654-f001:**
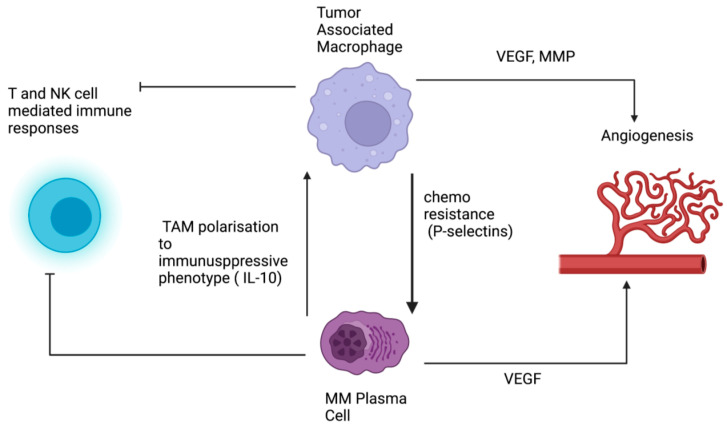
The interaction between multiple myeloma (MM) plasma cells and tumor-associated macrophages (TAM). Plasma cells promote polarization of TAMs towards an immunosuppressive phenotype, while TAMs confer chemoresistance properties to MM cells via P-selectins. Both TAMs and MM cells promote angiogenesis and suppress T-cell- and NK-cell-mediated anti-tumor immunity. NK = NK cell, VEGF = vascular endothelial growth factor, IL-10 = interleukin 10, MMP = matrix metalloprotease.

**Figure 2 cancers-14-05654-f002:**
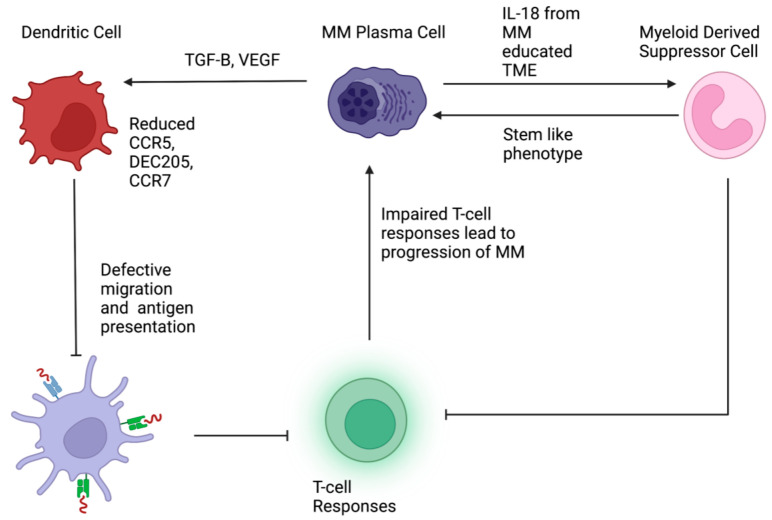
Interaction between multiple myeloma (MM) cells, dendritic cells (DC), and myeloid derived suppressor cells (MDSC). MM cells produce cytokines and growth factors, which lead to impaired differentiation, migration, and antigen presentation by DCs. Dysfunctional DCs lead to defective T cell activation. Overproduction of interleukin 18 (IL-18) in the MM tumor microenvironment drives MDSCS, which in turn suppress T cell responses. MDSCS also confer stem-like properties to MM cells increasing their proliferative capacity.

**Table 1 cancers-14-05654-t001:** Comparison of the characteristics of TAMs in solid tumors and MM; NR = not reported, PFS = progression-free survival, RFS = relapse-free survival, DSS = disease-specific survival, DMFS = distant-metastasis-free survival, and OS= overall survival.

Malignancy	TAM Phenotype	Biological/Clinical Implications	References
Non-small cell Lung Cancer	CD68 bright, CD163 bright	Anti-tumor function, improved survival	Ma 2010 [28], Rakaee 2019 [29]
Bladder	Galectin-9, CD68	Poor OS and RFS	Qi 2019 [30]
Head/Neck Squamous cell Cancer	CD163	Promote tumor progression leading to poor OS and PFS	Troiano 2019 [31]
Breast Cancer	(a) CD68 (b) CD163 (c) CD204	(a) Reduced OS, increased tumor stage and size (b) Reduced RFS and DSS (c) Poor OS, RFS and DMFS	(a) Parthia 2019 [32] (b) Klingen 2017 [33] (c) He 2019 [34]
Colorectal Cancer	(a) CD68, CD163, (b) Wnt5a (c) NOS2	(a) Counter the aggressive tumor budding phenotype, Counter cancer cell invasion (b) Reduced RFS, OS and Higher TNM stage (c) Increased RFS, Improved survival in a stage dependent manner	(a) Koezler 2016 [35] (b) Liu 2020 [36] (c) Edin 2012 [37]
Hepatocellular Carcinoma	Oncofetal TAM phenotype	Oncofetal macrophages influenced T-cell function	Sharma 2020 [27]
Multiple Myeloma	CD68, CD163	Pro-tumor function, reduced PFS/OS	Panchabhai 2016 [38], Suyani 2013 [39]

**Table 2 cancers-14-05654-t002:** Clinical studies implicating tumor-associated macrophages (TAM) as an adverse prognostic factor in MM; NR = not reported, PFS= progression-free survival, OS = overall survival RR = relapse rate. IHC = immunohistochemistry, FCM = flow cytometry, PI = proteasome inhibitor.

Technique for Identification of TAMs and Relevant Markers	Number of Patients	Treatment	Measure of Inferior Outcome	References
FCM and IHC (CD163)	10 FCM, 131 IHC	NR	OS	Panchabhai 2016 [38]
IHC (CD68 and CD163, iNOS)	240	12% PI, 67% IMID	RR, PFS, OS	Chen 2017 [60]
FCM (CD68, CCR2 for M1)	34	PI	OS (for low M1)	Beyar-Katz 2019 [64]
IHC (CD163, iNOS)	240	NR	OS/PFS	Wu 2015 [61]
IHC CD163	198	PI	PFS, OS, RR	Wang 2019 [62]
IHC (CD68, CD163)	68	-	OS	Suyani 2013 [39]
IHC (CD68)	136	NR	RR, PFS	Yu ASH 2012 [63]
Soluble CD206	104	NR	OS	Andersen 2015 [67]

**Table 3 cancers-14-05654-t003:** Studies evaluating the CD47:SIRP alpha axis in multiple myeloma (MM). MGUS = monoclonal gammopathy of undetermined significance. VAD = vincristine, doxorubicin, dexamethasone; ASCT = autologous stem cell transplant; Dara = daratumumab; PFS = progression-free survival; OS = overall survival; IHC = immunohistochemistry; FCM = flow cytometry; BM = bone marrow.

Study	Diagnosis, Treatment	Sample Size	Analysis Platform	Findings
Rendtlew et al. BJH 2007 [71]	MM and MGUS	171 MM patients and 18 MGUS patients	FCM	No OS difference. Lower CD47 expression in EMM, than BM MM
Sun Cancers 2020 [54]	MGUS 44, MM 559	MGUS 44, MM 559	GEP data	Higher CD47expression in MM compared to MGUS
Rastgoo Haematologica 2020 [72]	MM (VAD and ASCT)	74 newly diagnosed	IHC	Inferior PFS and OS
Storti BJH 2020 [74]	Newly dx and relapsed MM	11 newly diagnosed 13 relapsed	Ex vivo testing of dara on primary MM cells	High CD14/CD138 ratio was predictive of response to dara in vitro. CD16 monocytes are required for in vitro anti CD47 blockade mediated killing of MM cells.
Kambhampathi ASCO 2020 [73]	Relapsed MM BM trephine pre- and post-dara	11 relapsed	IHC, H scoring for CD47 and CD68 on BM trephine.	CD47 expression changed from surface to cytoplasmic at relapse. No change in CD68 or C47 expression overall at relapse.

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
