# Peer review of "Tumor-Associated Macrophages and Related Myelomonocytic Cells in the Tumor Microenvironment of Multiple Myeloma"

_cancers, 2022, doi:10.3390/cancers14225654_

Round 1

Reviewer 1 Report

This review is interesting, but I think the quality is a little low. First, the authors should describe the TAM overall function. Then, the authors should compare the TAM characteristics between other tumors and MM. The table indicating the comparison should be prepared. In the current version, the manuscript cannot be accepted. Taken together, major revision should be made before re-submission. The manuscript can be re-considered only when all the comments are responded.

1. Introduction

The authors should describe the TAM's overall function and application in vitro. I recommend the papers be quoted.

Progression  Journal of Biomedical Science volume 26, Article number: 78 (2019

Immunity  https://doi.org/10.3389/fimmu.2020.583084

Invasion or metastasis Journal of Hematology & Oncology volume 12, Article number: 76 (2019)

3D (Review and research)

Cancers 202012(10), 2754

Tissue Eng. Part A, 26, 2020, 1272-1282. https://doi.org/10.1089/ten.tea.2020.0095

EVs

https://doi.org/10.1016/j.neo.2018.06.004

https://doi.org/10.1002/jcp.29784

Author Response

We are grateful for the valuable suggestions. We have included more details in the introduction regarding the role of TAMs in tumor progression, metastasis and immune modulation. We have also discussed the role of exosomal miRNA transfer as suggested. We have included a discussion on ex vivo drug testing and 3D tumor models in paragraph 3 of section 9. We have included a new table ( table 1) describing the characteristics and biological role of TAMs in solid tumors and comparing this to myeloma.

Reviewer 2 Report

The authors descript and discuss the tumor microenvironment of Multiple Myeloma about the relationship between Macrophages, MDSC, and DCs. And further, discuss the axis of CD47 and SIRPα. Before publication, there are some errors to be corrected in the article.

1. The style of references should be according to the rules of the journal "Cancers".

2. Please add serial numbers to the subtitles.

3. Some double space in the sentences, please delete it.

4. Line 138. The authors thought that TAMs could promote chemo-resistance in MM. In vitro studies, some MM cell lines become drug resistant without TAMs. It seems to need to descript more detail.

5. Cancer stem cells and stroma cells also have essential roles in the TAMs. Please add the relationship and descript in the article.

6. Please cite the references in Table 1.

7. Please cite the clinical trials number or references in Table 2.

Author Response

We are grateful to the reviewer for the comments.

  1. The style of references should be according to the rules of the journal "Cancers".

Authors Reply: manuscript has been modified to MDPI reference style which is the reference style for “Cancers”

  1. Please add serial numbers to the subtitles.

Authors Reply: the subtitle for each section is numbered.

  1. Some double space in the sentences, please delete it.

Authors Reply: Double spaces in the sentences have been removed.

  1. Line 138. The authors thought that TAMs could promote chemo-resistance in MM. In vitro studies, some MM cell lines become drug resistant without TAMs. It seems to need to descript more detail.

Authors Reply: Thank you for the comment, we agree that the role of TAMs in MM drug resistance is incompletely understood, we have elaborated on this in section 2 paragraph 4.

  1. Cancer stem cells and stroma cells also have essential roles in the TAMs. Please add the relationship and descript in the article

Authors Reply: Thank you for the comment. We have expanded upon the relationship cancer stem cells and stromal cells have with TAMs. In the introduction, paragraph 3.

  1. Please cite the references in Table 1

Authors Reply: References have been added to  tables 1 and 2.

  1. Please cite the clinical trials number or references in Table 2.

Authors Reply: References have been added to Table 2.

Round 2

Reviewer 1 Report

I recommend the publication.